# Research on LMS and KPIs for Learning Analysis in Education

Milena Krumova

Faculty of German Engineering Education and Industrial Management, Technical University of Sofia,
1000 Sofia, Bulgaria; mkrumova@tu-sofia.bg

**Abstract:** Learning Analysis (LA) trends show that KPIs used for better understanding and investigation of the learning process are diverse and often depend on the goals of the analyses. One of the most crucial questions for the implementation of LA is the availability of data. Currently, each education organization uses some type of Learning Management System (LMS), thus gathering, storing, and managing different data about the learning process. The initial point of this research is the fact that the time framework of the learning process is predefined, and that learning process analyses must be done according to this limit because each learning activity takes time to be performed. This research conducts an overview of the LMS and proposes KPI clusters for measuring and analyzing learning processes regarding (i) learning engagement, (ii) learning community building and management, and (iii) knowledge no-boundaries. It is structured into three parts. Based on the research methods used (a literature review, desk research, and experiments), (1) a review of LMS is conducted, then (2) the learning analysis is presented, (3) KPI analyses are done, and, finally, a new KPI model for LA is proposed, which includes 22 KPIs grouped into 4 Clusters. In the conclusion, the advantages of the model are explained along with the scientific contribution of the research. The next research steps, regarding the implementation of the model in the real classroom, are presented at the end.

**Keywords:** learning analysis; LMS; data; KPIs; learning process

## 1. Introduction

Many innovations have been created and implemented into the field of education during the last several years, such as technology-based learning, the usage growth of the web 2.0 tools and applications for learning (learning 3.0/4.0), novelty pedagogy approaches, m-learning models, changes in digital learning content, etc. Educators are continuously looking for new models and innovations to improve and adapt the learning process to learners' generational needs [1,2], in order to improve efficiency and be more effective in the achievement of learning objectives. Time efficiency is a Key Performance Indicator (KPI), which can be measured based on data about the learning process. It can be measured in different contexts and by using different goals. One example is presented in a study that measured KPI students' engagement within the LMS on various course components. The analyses use the amount of times students view/complete various course components in the LMS and how these views/completions compare with other students [3]. Another example is the analysis of the KPI course insights engagement within LMS. Student engagement is measured by the average activity pattern per student, which is based on the student being active in LMS Blackboard or edX Edge (i.e., clicking or watching a video) [4]. The measurement of the education and specifically the measurement of the learning process using KPIs is related to the time framework due to the "Credit hours" and "Time hours." It can be said that the time framework is also a precondition for learning to occur, especially for the traditional classroom model, where the learning outcome is time-dependent, which differs from the Competency-Based model and STEM Problem-Based model, where the learning outcome is not time-dependent.

There are huge differences in Credit hours from one curriculum to the other. These differences are a result of the national education standards, the education system, knowl-

edge domains, the education level, school year length, the courses in the curriculum, etc. In the common traditional education model, the school year has two semesters; however, the number of weeks within each semester can vary. For example, the school year of the Technical university of Sofia has two semesters, each one 13 weeks, the school year of the technology school "Electronic Systems" has two semesters, and each one has 18 weeks. Each course within a semester of a curriculum depends on the knowledge domain and includes predefined hours for teaching-learning—lectures, seminars, laboratory work, project works, etc. Courses credit hours and time required also vary in such a curriculum time framework. In order to achieve the learning objectives, the lecturers and teachers must also take into account the time regarding the learner's ability to perform different learning activities (Figure 1). For example, a normal rate for learning is 100–200 words per minute (wpm), and for comprehension, it is 200–400 wpm. The speed of reading, when it is done normally, is usually at a rate of around 400–700 wpm [5]. Learners' abilities to perform different learning activities, such as analyzing, discussing, observing, presenting, summarizing, etc., according to Bloom's taxonomy [6], also vary regarding the length of time. This study tries to answer the research question "What KPIs for learning process analyses should be used based on the data available from LMS, and out of LMS, in support of better time efficiency?".

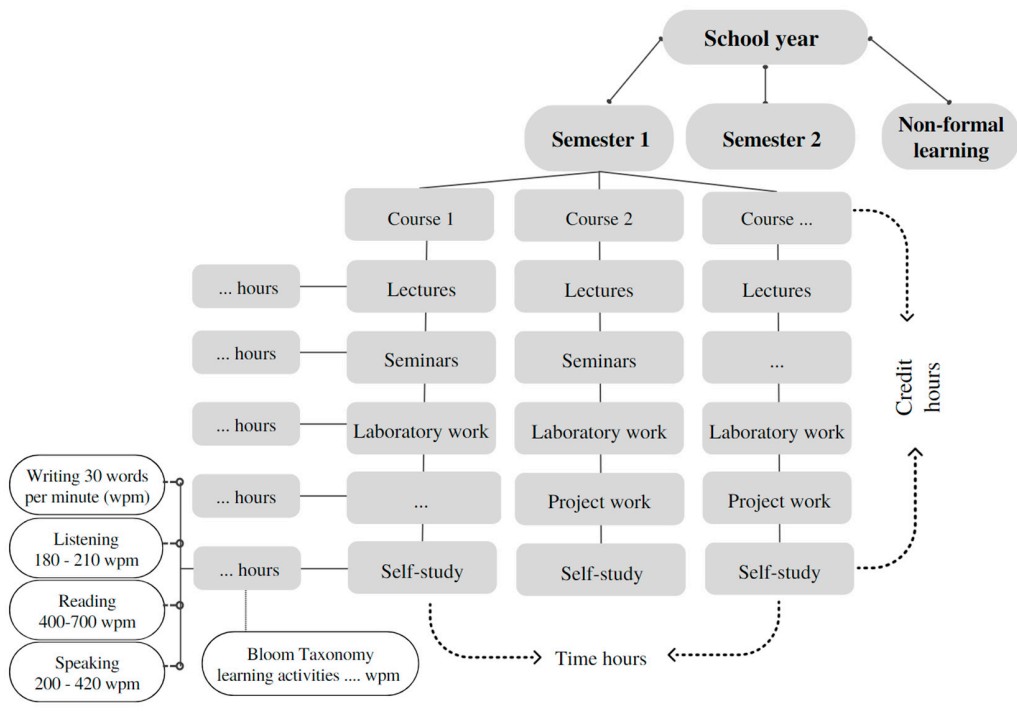

**Figure 1.** School year—Credit hours and Time hours map.

## 2. Learning Management Systems

Since the first Online Degrees have been designed [7], LMSs have continued to evolve. LMS includes software technologies and applications to support the management of digital training courses and content. LMSs simplify the administration and management of the learning process by supporting the instructors in learning design and creating more engaging courses, leading to learners' more-effective participation, use and re-use of digital learning materials, and knowledge retention through a single platform. The use of LMS is studied in a diversity of contexts. Research has examined how the implementation of case method learning is carried out using LMS media [8]. Another study shows the importance of transforming the e-learning landscape through achieving LMS system interoperability, open curriculum, e-portfolio management, and online teaching and learning assistance [9]. Other research has analyzed the trend of LMS development [10], and analyzed the pedagog-

ical aspects of LMS Edulogy [11]. Similarly, another study introduced a framework, along with an evaluation of the functionality of the LMS, for measuring the intended learning outcomes [12].

LMS [13] is a strategic tool for each educational organization and one of its biggest advantages is a deeper understanding of the learning process analyses. The strategic decision for the use of an LMS is based on the organizational vision, mission, and objectives. A good LMS fosters a learner-focused approach by supporting the education organization in three key areas: Learning, Progress, and Achievement [14]. The best LMS for education work is also the best for present-moment data management, however, they are also used for planning, forecasting, prevention, and analysis [15]. The criteria that are taken into account during the process for choosing, implementing, or changing LMS [16] are diverse: numbers of users; deployment model; features and functionality; SCORM compliance; pricing model; ability to scale; integration to other systems; etc.

During the learners' path, and the data gathering process, LMS is a data source for learning analysis and measuring using KPIs. Another source can be Massive Open Online Courses (MOOC) platforms [17], web-based learning tools (web 2.0/3.0), and audio devices such as microphones, cameras, keyboards, social networks, and others (Figure 2).

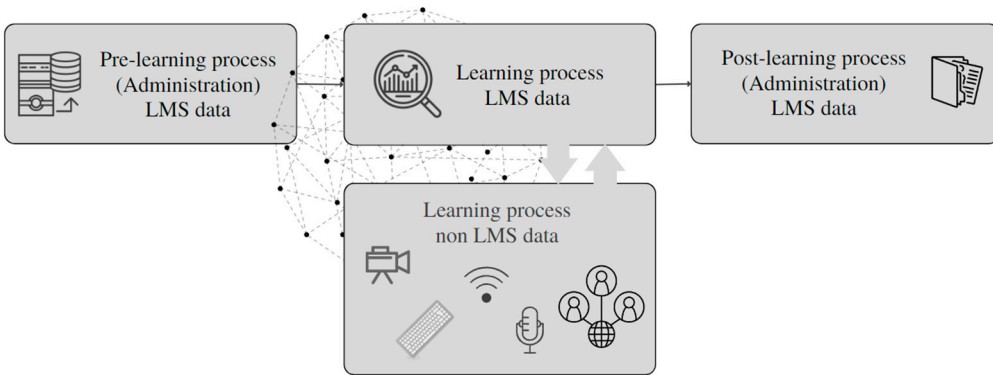

**Figure 2.** LMS Data and Non-LMS data gathering.

According to IMS Global (In 2022, IMS became 1EdTech), successful technologies (LMS) are those that make teaching and learning easier and more productive. IMS enables a plug and play architecture and ecosystem that provides a foundation on which innovative products are rapidly deployed and work together [18]. A key question related to LMS is the data standardized format. This issue was addressed in 2004 with the introduction of the Sharable Content Object Reference Model or SCORM file. This standard format which specifies parameters on content packaging and metadata has now become a foundation of modern LMS [19]. In Table 1, the technical work and technical focus of 1EdTech are presented.

**Table 1.** 1EdTech Technical work and technical focus [18].

| Technical Work | Technical Focus |
|---|---|
| Accessibility | Accessibility, inclusive design, and personalization of online learning resources to meet the needs of all users/learners. |
| Caliper Analytics | Consistently capturing and presenting measures of learning activity and defining a common language for labeling learning data, with a standard way of measuring learning activities and effectiveness enabling designers and providers of curriculum to measure, compare and improve quality. |
| Common Cartridge | Organization, publishing, distribution, delivery, and search of a wide variety of collections of digital learning content, applications used as the basis for or in support of online learning of any type. |

**Table 1.** *Cont.*

| Technical Work | Technical Focus |
|---|---|
| Competencies and Academic Standards Exchange | Exchange and management of information about learning standards and/or competencies in a consistent and referenceable way including machine-readable statements of what the learner will know and be able to do, explanations of relationships between standard sets and/or among individual standards or courses where applicable, and guides listing specific criteria for grading or scoring academic papers, projects, or tests. |
| Comprehensive Learner Record Standard | A comprehensive digital learner record applicable to education and workforce learning. The 1EdTech CLR Standard supports competency-based education, co-curricular and extra-curricular skills and achievements, employer-based learning, and other learning experiences in a verifiable, portable, and interoperable format. |
| Data Privacy | Vetting educational applications, to ensure that a minimum standard of privacy and security is met, provides assurance that the information gathered by these educational applications is being used responsibly, thereby certifying that institutions are using TrustEd Apps. |
| EPUB for Education | Establishing a globally interoperable, accessible, open ecosystem for e-Textbooks and other Digital Learning Materials via EPUB3, Educational Sector Standards and the Open Web Platform. |
| Learning Tools Interoperability | Integrating rich learning applications (often remotely hosted and provided through third-party services) with platforms like learning management systems, portals, learning object repositories, or other educational environments. |
| Learning Tools Interoperability Resource Search | Searching digital repositories for a set of resources using various attributes of resources and returning full metadata about the resources to the learning tool. |
| OneRoster/Learning Information Services/Edu-API | Exchange and synchronization of roster information and grades which focuses on people, memberships, courses, and outcomes. |
| Open Badges | Capturing learner achievements that are verifiable, portable, and interoperable through Open Badges and in the future, learning pathways and Blockchain extensions. |
| OpenVideo | Creation of video capture metadata to enable information from any video to be shared and searched. |
| QTI 3 and Assessment | Exchange of item, test and results data between authoring tools, item banks, test construction tools, learning systems and assessment delivery systems, including accessible assessments. |

Among the key characteristics of LMS are: flexible training options; social learning and learner engagement; authoring tools; mobile learning functionalities; talent management; gamification; SCORM; video conferencing; Microlearning; ePortfolio; etc. Regarding the deployment model, LMSs can be grouped into Open Source Software and Web-Based Software. For instance, the Capterra platform includes 1185 LMS [20]. Some of the most popular OSS LMS are Moodle, ILIAS [21], and Chamilo [22], and among the best web-based LMSs are Stile, Google Classroom, and Perusall online social learning platform. The highest-rated are: D2L Brightspace; Tovuti; TalentLMS; Docebo [23].

### 3. Learning Analytics

Learning analytics (LA) is used in Technology Enhanced Learning (TEL) to track learning progress and empower educators and learners to make well-informed data-driven decisions. LA consists of analyzing educational data to enhance the learning experience. For example, it can determine the time a student spends on a specific activity and the number of visits to it [17]. LMS LA functionalities allow education organizations to continuously improve their learning process efficiency. They can help to monitor the following KPIs:

- Identify at-risk students and increase retention;

- Predict the likelihood of college readiness;
- Prepare students for further study or their chosen profession;
- Shorten the time to graduation;
- Improve student engagement and satisfaction;
- Monitor instructor effectiveness;
- Boost the reliability of assessments used to evaluate student success;
- Automate reward schedules and introduce game mechanics at scale;
- Gauge program performance and determine areas for improvement;
- Understand how learning technology is being used;
- Analyze data for benchmarking and research;
- Measure the effectiveness of new learning strategies.

*3.1. Moodle*

Moodle is a free customisable OSS LMS used by over 300 million people in over 100 languages worldwide. The feature-rich platform is secure and scalable, integrates seamlessly with third-party platforms and plugins, and allows teachers to create engaging, accessible, and active learning experiences. Learning analytics are software algorithms that are used to predict or detect unknown aspects of the learning process, based on historical data and current behavior. Moodle can be used with an LA system, including an API for the collection and combination of learning analytics data and a number of interfaces for presenting this information (Figure 3) [24].

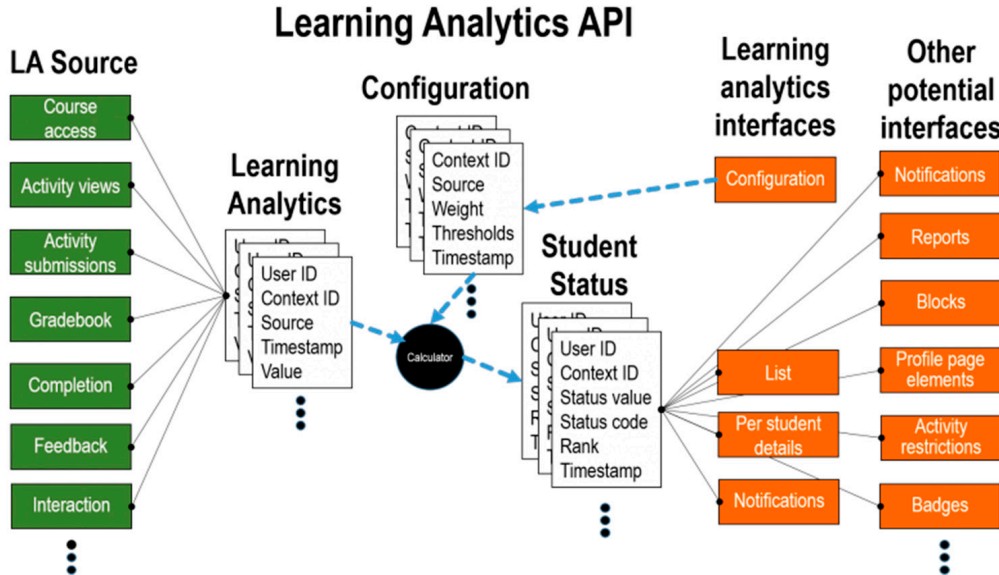

**Figure 3.** Moodle—Learning Analytics data gathering API [24].

There are four main categories of LA: descriptive (what happened?); predictive (what will happen next?); diagnostic (why did it happen?); prescriptive (do this to improve). Moodle provides a variety of built-in reports based on log data; however, they are primarily descriptive in nature—they tell participants what happened, but not why, and they do not predict outcomes or advise participants how to improve outcomes. Moodle includes three types of reports:

- Course activity reports

A course activity report, showing the number of views for each activity and resource (and any related blog entries), can be viewed by managers, teachers, and non-editing teachers (and any other users with the capability report/outline).

- Individual activity reports

If activity reports are enabled for a course in the course settings, each course participant can access reports of their contributions, such as forum posts or assignment submissions, logs, and a statistics report.

- Complete report

The Complete report allows instructors to view a detailed list of an individual student's last log and activity in the Activities and resources in the Moodle course, including detailed contributions to any of the various types of course activities. The activities and resources are displayed in the same order as they are on the main course page, it might resemble a student's portfolio for a specific course.

### 3.2. Stile LMS

Stile is a web-based LMS that is Australia's leading secondary science program, used by one in three secondary schools. Lessons, exams, and other activities in Stile can be viewed in three modes. These modes correspond to the workflow before, during, and after class and keep lesson creation tools separate from the teaching tools. Teach Mode is an in-class presentation tool designed to help deliver evidence-based teaching strategies with Stile. Analyze Mode provides a view of students' learning after a lesson. It shows who has started or submitted a lesson and an overview of how they have progressed through it. Stile also allows for running a scientific investigation.

Each subject in Stile has its own Mark book, which shows a grid view of every student and every lesson within that subject. It is the quickest way to see an overview of students' activities, and can easily narrow in on the details. Once the students have completed work, the Mark book helps to track and report on student achievement by keeping all of the information needed in one place (Figure 4). Patterns and trends can be evaluated at a whole class level, or the educator can track a specific student's progress and request resubmission of work [25].

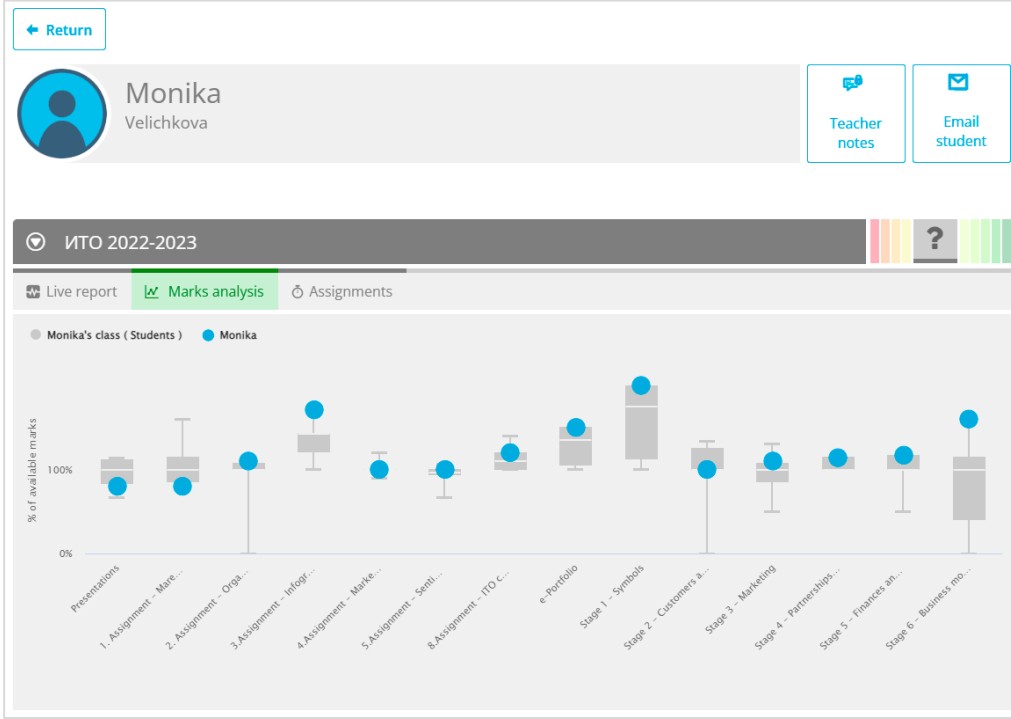

**Figure 4.** Stile dashboard "Individual Mark book" (Screenshot of a student portfolio).

### 3.3. Perusall

Perusall is an online social learning platform. Perusall facilitates a deep reading experience in a social environment by grouping students into small learning communities for each reading assignment (i.e., textbook chapter, article). Students can directly annotate on the document by posting comments or questions, which will be graded by Perusall's Artificial Intelligence for quantity, thoroughness, and quality. Instructors can review various engagement analytics (including the most up-voted questions and most popular comments) and interact themselves, including providing critical thinking prompts throughout the text [26]. Perusall helps students prepare for class by engaging with materials and each other before a lecture. More time can be spent on higher-order skills (analyzing, creating, evaluating) during class time through active learning activities. The tool turns tasks, such as pre-readings, that are usually completed individually as an isolated activity, into a social experience. LA Perusall includes (Figure 5):

- Grade distribution;
- Annotation submission time heat map;
- Page view report
- Student activity report
- Overall assignment progress
  - Completed with maximum score
  - Completed but not maximum score
  - Some work submitted
  - No work submitted

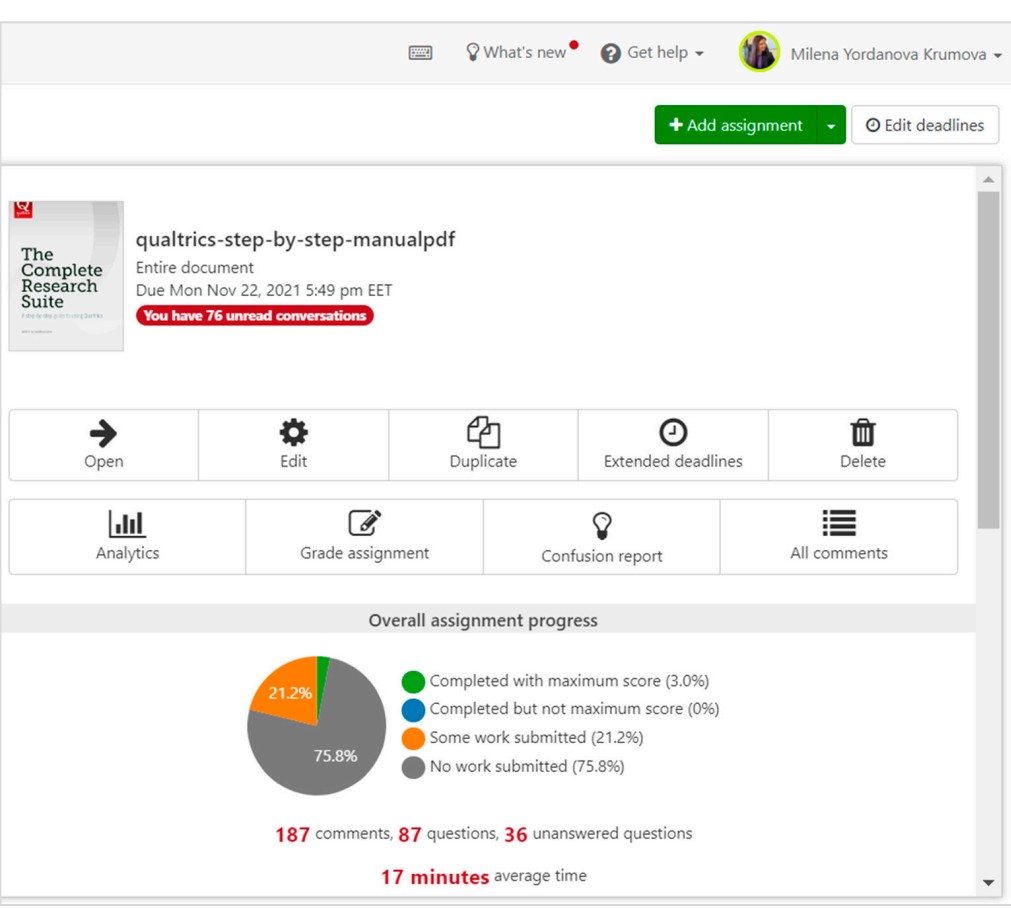

**Figure 5.** Perusall dashboard "Overall assignment progress" (Screenshot of class work).

### 3.4. Mural

Mural is a collaborative tool that allows teamwork in flexible, inclusive collaboration spaces. It makes it easy to embrace visual collaboration for education for schools and universities of all sizes. In Figure 6, a LA dashboard is shown.

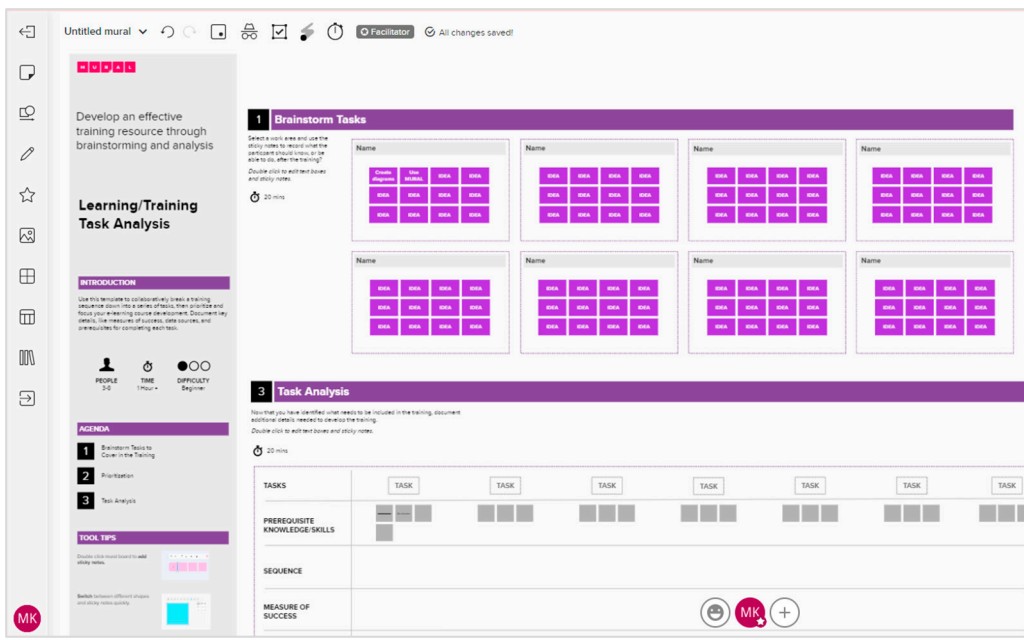

**Figure 6.** Mural dashboard "Task analysis."

Mural allows participants to both learn and test key practices in one space using hundreds of templates for collaboration: study guide template, meta storyboarding template; E-learning storyboard template; curriculum development template; etc. Mural is an efficient non-LMS source for learning analytics [27]. Mural is applicable to different learning activities, such as project work, brainstorming activities, mind mapping, education event organizing, and team building class activities.

## 4. LA Key Performance Indicators (KPIs) Novelty 4-Clusters Model

A key performance indicator (KPI) is a type of performance measurement that helps to understand how an organization, department, or institution is performing and allows for an understanding of the direction in which the strategy is headed. KPIs are identified and used by the authors on a diversity of levels and frameworks: international, national, and organizational. According to research, there are 28 critical education KPIs, grouped into nine clusters: Academia; Finances; Ratios; Curriculum; Faculty; Facilities; Technology; Transportation; Housing [28]. The KPI mega library includes 36,000 KPIs grouped into organization, government, and international levels. Some examples used for education have been grouped into the following categories: administration; assessment; career service; disability services; engagement and partnership; enrollment; facilities and equipment; faculty and staff; financial; financial aid; international learning; library; participation; public relations; quality assurance; register; research; retention; student achievement; student body; student learning; student support [29]. Some of the data used to analyze KPIs are not related to the LMS but are available from other sources, such as administrative systems and financial systems, as well as other research strictly focused on KPIs totally based on educational LMS. An example is one study which had the purpose of studying the establishment of University e-learning knowledge systems. The research developed an e-learning knowledge system consisting of two dimensions of KPIs, "organization" and "teaching team," consisting of 84 KPIs in total [30]. Similarly, another study used KPIs to analyze students' performance in regard to changes in teaching methodology [31]. An excellent

example of the effective usage of KPIs is Zoomi AI. It is an Artificial Intelligence tool, which discovers patterns of learning and behavior to identify and understand each unique learning style based on the interaction within LMS. Based on KPIs, "Learner DNA" is used to make critical decisions that save time and resources. Zoomi uses over 250 algorithms [32]. Another example is eight metrics that are used to measure learner engagement: Sign-up rates; Completion and drop-out rates; Weekly and monthly active user numbers; Mandatory vs. voluntary course participation; Learning time spent; Community interactions; Feedback survey results; Percentage of employees using their new skills [33]. Similarly, 15 significant KPIs are integrated into Moodle analytics [34].

✔ Log in rate—How frequently is the user logging in?
✔ Enrolment/Sign up rate—How well is the latest course/s doing?
✔ Course status—What is the learners' learning/training status?
✔ Course completion rate—Are the learners finishing off courses or leaving away midway?
✔ Average time-spent rate on the course (or Train up rate)—How much average time do the learners spend on their course (s)?
✔ Course rating—How much is the learner satisfied with a training course?
✔ Learner/Trainer feedback rate—What's learner's/trainer's say on their learning/training respectively?

Another study stressed the attention paid to the course activity and discussion forum interactions:

- *Course activity:* When are students active on the course website? Which students are most active? Are there consistent usage patterns? How does activity on the site change over time?
- *Discussion forum interactions*: Which students are talking to one another? Which students are not engaged? Who are the connectors?

Student-LMS Activities are measured by: actions per day; actions per week; total actions; actions per weekday; actions per hour [35].

It can be concluded that some of the characteristics of the KPIs for the learning process are studied in the context of the strategic development of the organization, while others are studied regarding one or more learning process elements, such as the use of technologies, teaching methods, and LMS interaction. The Gap that was identified based on the analyzed literature, and studied LMS and LA tools, as well as real classroom experiments, is the lack of:

- KPIs for capturing and analysis of the specific learning activities and the time necessary for their performance (reading time, resuming time, searching time, solving problems time, etc.; Bloom's taxonomy activities);
- KPIs for studying the impact of the knowledge sources' availability in a digital and non-digital format on student performance;
- KPIs for analyzing the impact of the digital nature of the students in regard to the theory of Connectivism and students' need to be connected in a learning community;
- KPIs related to the link between the learning objectives, set at the beginning of the learning process and the learning outcome achievements, are presented in the students' e-portfolio.

Based on the analyzed literature resources, desk research, and LMS experiments, this research proposes a novelty LA KPIs Four-Clusters model. It is shown in Figure 7. The model can be implemented per course and the analyses can measure the trends per year, per semester, per week, per day, and/or per hour. The model includes 22 KPIs grouped into four clusters. The data sources are LMS and external data systems, such as web 2.0/3.0 tools, social media, etc. Some of the data are gathered through feedback analyses. In the following, Table 2, KPIs are presented alongside the sources of data for their measurement.

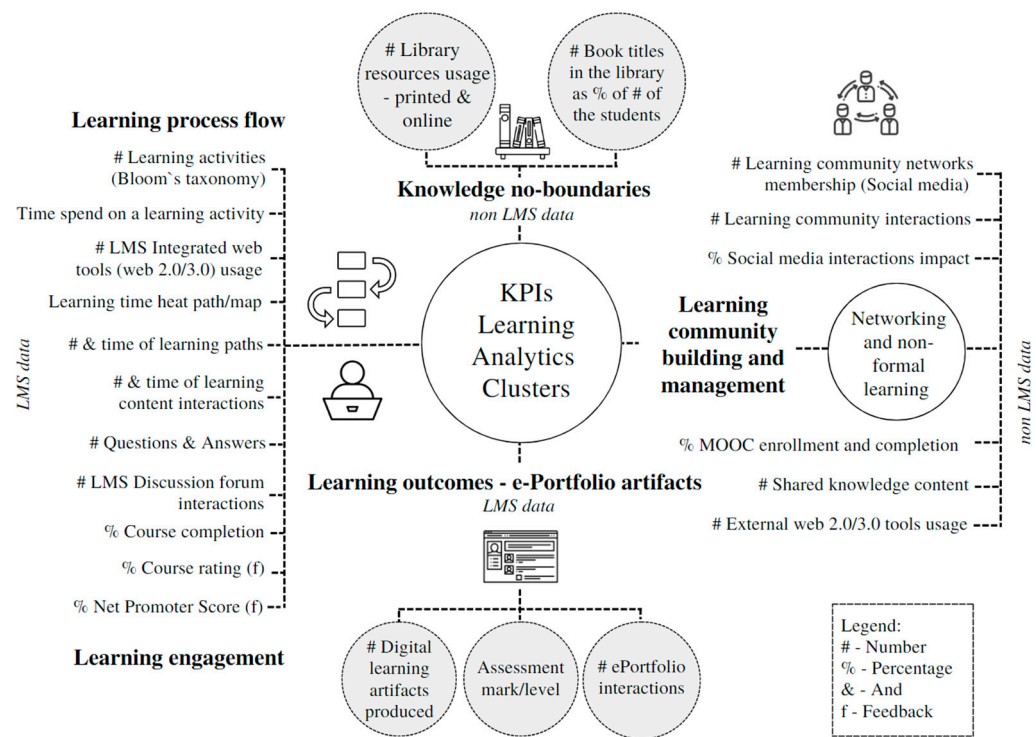

**Figure 7.** Learning Analytics KPIs Four-Clusters model.

**Table 2.** Learning Analytics KPIs—indicators explanations and data source.

| KPIs Clusters | KPI | Explanation | Data Source |
|---|---|---|---|
| Learning process flow | # Learning activities (Bloom's taxonomy) | Reading; Resuming; Comparing; Creating; etc. How many and what are the learning activities included in the learning process performed by the student? | LMS |
| | Time spend on a learning activity | Reading—3 min. Making analyse—25 min. | |
| | # LMS Integrated web tools (web 2.0/3.0) usage | Web 2.0 tools—Canva, Pixton, Mural, MindMeister, Tinkercad, etc. How many tools are used by the student to perform different learning tasks? | |
| | Learning time heat path/map | Learning activities timeline. When do the student learn during the day? (For ex. at 5 pm.) | |
| | # & time of learning paths | What is the period/time from the beginning until the final part of the learning process? How many are the learning paths? | |
| | # & time of learning content interactions | How many times have been read the module? How many times have been watched the video? etc. | |
| | # Questions & Answers | How many questions are raised to the lecturer and the peers? How many answers are given to peers? | |
| | # LMS Discussion forum interactions | How intensive (often) the student uses the discussion LMS forum during the learning. | |
| | % Course completion | Does the student do all the assignments? How many are not completed? | Feedback |
| | % Course rating (f) | How the student rates the course? Good-Average-Poor | |
| | % Net Promoter Score (f) | Is the student willing to and recommend the course? | |

**Table 2.** *Cont.*

| KPIs Clusters | KPI | Explanation | Data Source |
|---|---|---|---|
| Knowledge no-boundaries | # Library resources usage—printed & online | How many resources are used by the student? | Library & Feedback |
| | # Book titles in the library as % of # of the students | How many books (per course titles) per student are available in the library? | |
| Learning outcomes—e-Portfolio | # Digital learning artifacts produced | A number of artifacts created—resumes; multimedia; screencasts; avatars; etc. by the student. | LMS |
| | Assessment mark/level | Final exam score/mark | |
| | # ePortfolio interactions | How often does the student interact with ePortfolio to upload and improve the learning process artifacts? | |
| Learning community building and management | # Learning community networks membership (Social media) | How many social media communities do the student participate in during the learning process? Facebook groups; LinkedIn groups; etc. | Social media analyses—open data |
| | # Learning community interactions | How many times does the student interact with the learning community? High-Medium-Low | |
| | % Social media interactions impact | How impactful is the student's interaction with the learning community? High-Medium-Low | |
| | % MOOC enrollment and completion | Is the student enrolled in MOOC? | Feedback |
| | # Shared knowledge content | Number of shared knowledge (eBooks, Guidelines, etc.) with peers out of LMS. | |
| | # External web 2.0/3.0 tools usage | Number of tools usage (Screen-O-Matic, Plotagon, etc.) by the student. | |

Legend: #—Number; %—Percentage; f—Feedback.

The model corresponds to the SMART principle that everything which can be measured can be analyzed and improved, including the learning process. SMART approach (Specific, Measurable, Achievable, Reasonable, and Time Related) for strategic development in education is among the most important approaches, which can give the educational organization a clear picture for the improvement of the learning process and the implementation of innovations.

## 5. Conclusions

The learning process is one of the many objects within the education system which have many possibilities for improvement, and the sources for improvements are diverse. On one hand, the digitalization processes in education, the digital learners' needs, along with the competition among the LMS vendors, has boosted the use of LMS by educational organizations. On the other hand, during the last two decades, and mostly during the COVID-19 pandemic, the number of LMSs has increased enormously. This opened many new possibilities for learning analysis using KPIs and learning process discoveries, as well as new models for learning process assessment design. This research proposes Learning Analytics KPIs Four-Clusters model. The model consists of four clusters of KPIs measurement: (1) learning process flow and learning engagement, (2) learning outcomes—ePortfolio artifacts, (3) knowledge no-boundaries, and (4) learning community building and management. The main distinguishing characteristic of the KPIs model over the other models for learning process analysis is the uniqueness of the possibilities much more precisely assess all learning activities regarding invested time in relation to the achievement of the learning outcomes presented in the e-portfolio. In this way, the monitoring of such activities, which bring a positive impact on the learning outcomes, can be intensified, and those having a

negative impact can be recommended not to be implemented into the learning process design. The model differs in its specifics, and on account of its correspondence to the digital nature and connectivity of 21st-century learners, all KPIs are based on the use of digital tools or digital interaction. The advantage of the model is its potential to be implemented in different periods of time, as well as different levels of complexity on the basis of how many activities are included in the learning process within LMS. Due to the many possibilities for the use of Bloom's Taxonomy, there are many possibilities to be analyzed. Another advantage of the Learning Analytics KPIs Four-Clusters model is the access, use, and interaction with knowledge content available from the library, which can show if a correlation between the diversity of knowledge content and learning activity types exists. These and other questions of deeper understanding are going to be an integral part of the next research steps, e.g., the implementation of the model into a real learning environment, while more light is shed on the weight of each KPI and the mathematical model for the evaluation and analyses. The following research questions "Are there priority KPIs? How often should the learning analysis be conducted?" are going to be analyzed and answered as well.

**Funding:** This research received no external funding.

**Data Availability Statement:** No new data were created or analyzed in this study. Data sharing is not applicable to this article.

**Conflicts of Interest:** The author declares no conflict of interest. The funders had no role in the design of the study; in the collection, analyses, or interpretation of data; in the writing of the manuscript; or in the decision to publish the results.

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
