# Peer review of "Research on LMS and KPIs for Learning Analysis in Education"

_smartcities, doi:10.3390/smartcities6010029_

Round 1

Reviewer 1 Report

The paper aims to identify the kpi to be analyzed using data taken outside and inside an LMS to support time efficiency in learning. 

The introduction is clear and well structured and gives the reader the correct information to continue reading the article.
The cited literature is up-to-date and is consistent with the research presented.

I find some weakness for the article in paragraphs 2 and 3.

An overview of the main lms is given and the learning analytics process is broadly described. I think it is not necessary to go in deep with these arguments which are already consolidated and widely debated in the academic community. It is not necessary to be repetitive.

I suggest including a literature review citing similar research on KPIs, evaluating learning efficiency, and optimizing learning times.

Here are some examples of papers that could be cited:

https://www.mdpi.com/2076-3417/12/17/8438
https://www.frontiersin.org/articles/10.3389/feduc.2020.583562/full
https://ieeexplore.ieee.org/document/9582096
https://www.semanticscholar.org/paper/A-KPI-Based-Approach-to-Performance-Oriented-Wang/d53434755a23869a6d9e32a1ab973c753a79460f
https://ieeexplore.ieee.org/abstract/document/6745564
https://publications.eai.eu/index.php/el/article/view/1788

The paragraph 4, the most interesting and central one is very important to answering the research question, should be better cared for and improved. It is necessary to specify how the list of KPIs has been developed, if these have been tested or if it is foreseen to do so in future research (if a test has not been performed, I think the paper is more suitable for a conference).

Paragraph 4 needs to be significantly improved and extended.

Conclusions and future developments need to be improved on the basis of the changes that will be made in paragraph 4.

Author Response

Thank you so much for the review.

I believe, now, the manuscript is improved.

I agree with the review comments. Some of the mentioned recommendations are taken into account. Please, see the final version. The 3rd and 4th parts are improved. The proposed model is explained in more detail, while all 22 KPIs are presented in a table with explanations and data sources. The conclusion is elaborated in a new way and shows not only the scientific contribution but also the next research questions. Some additional sources are cited. The literature is checked to be in accordance with the style. 

Best wishes,

Milena Krumova

Reviewer 2 Report

Dear Author,

Learning analytics is an interesting topic. However, I propose some changes and improvements.

1. The title and keywords are clear and inform about the topic to be covered.

2. The abstract is good, but the last sentence "In the conclusion the next steps of the research are given", I consider that it does not provide significant data. I propose to improve this last part by providing real conclusions about the study carried out.

3. Sections 1, 2, 3 and 4 provide enough previous studies that are very up to date. Is Table 1 self-created or based on previous studies? This should be indicated.

4. The conclusions are poor and do not show a substantiated reflection of the results found. Furthermore, they are not based on the findings of previous studies. The limitations of the research are not stated. This section needs substantial improvement.

5. The references do not conform to the MDPI standard. They need to be improved.

I strongly recommend improving these aspects to give more value to the article presented.

Author Response

Thank you so much for the review.

I believe, now, the manuscript is improved.

I agree with the review comments. Some of the mentioned recommendations are taken into account. Please, see the final version. The 3rd and 4th parts are improved. The proposed model is explained in more detail, while all 22 KPIs are presented in a table with explanations and data sources. The conclusion is elaborated in a new way and shows not only the scientific contribution but also the next research questions. Some additional sources are cited. The literature is checked to be in accordance with the style. 

Best wishes,

Round 2

Reviewer 1 Report

The article was improved and it can be considered for publication. I think that the empirical study to test the KPIs is missing and that's the main flaw of the paper.